# Modeling and Simulation of Biogas Production in Full Scale with Time Series Analysis

**DOI:** 10.3390/microorganisms9020324

**Published:** 2021-02-05

**Authors:** Celina Dittmer, Johannes Krümpel, Andreas Lemmer

**Affiliations:** State Institute of Agricultural Engineering and Bioenergy, University of Hohenheim, Garbenstrasse 9, 70599 Stuttgart, Germany; j.kruempel@uni-hohenheim.de (J.K.); andreas.lemmer@uni-hohenheim.de (A.L.)

**Keywords:** demand-orientated, ADM1, self-learning, regression model, forecast, prediction

## Abstract

Future biogas plants must be able to produce biogas according to demand, which requires proactive feeding management. Therefore, the simulation of biogas production depending on the substrate supply is assumed. Most simulation models are based on the complex Anaerobic Digestion Model No. 1 (ADM1). The ADM1 includes a large number of parameters for all biochemical and physicochemical process steps, which have to be carefully adjusted to represent the conditions of a respective full-scale biogas plant. Due to a deficiency of reliable measurement technology and process monitoring, nearly none of these parameters are available for full-scale plants. The present research investigation shows a simulation model, which is based on the principle of time series analysis and uses only historical data of biogas formation and solid substrate supply, without differentiation of individual substrates. The results of an extensive evaluation of the model over 366 simulations with 48-h horizon show a mean absolute percentage error (MAPE) of 14–18%. The evaluation is based on two different digesters and demonstrated that the model is self-learning and automatically adaptable to the respective application, independent of the substrate’s composition.

## 1. Introduction

Worldwide, it is imperative that the energy supply based on renewable energy resources increases. In Germany, the Renewable Energy Act stipulates that 80% of the gross electricity supply must be covered by renewable energies by the year 2050 [1]. Due to low production costs and greenhouse emissions, the future power systems will be characterized by fluctuating feeds from wind turbines and photovoltaics (PV). Therefore, technologies to balance the divergence between energy supply and demand are gaining increasing relevance. Besides the adapted behavior of energy consumers (demand-side management) and the installation of new electricity storage capacities, a higher importance will be placed on load-flexible and modular energy production [2]. In this context, bioenergy, especially biogas plants, can play a crucial role. Compared to biogas plants designed for constant load, the demand-oriented, load-flexible use of biogas requires significantly larger combined heat and power units (CHP) and corresponding gas storage facilities [3]. However, on-site gas storages are limited for cost, safety, and licensing reasons [4]. Future biogas plants must therefore also be able to produce biogas according to demand, which in turn requires proactive feed management [5]. Such intelligent feeding systems can also reduce the construction costs of biogas plants, as the required gas storage capacities can be reduced by up to 65% [6].

Concepts for feeding management require the simulation of biogas production depending on the substrate supply [7]. Most simulation models are based on the complex Anaerobic Digestion Model No. 1 (ADM1), edited by [8]. Parameters of all biochemical process steps (hydrolysis, acidogenesis, acetogenesis, methanogenesis) as well as physicochemical processes (ion association/dissociation and gas-liquid transfer, etc.) are considered in this model [9]. According to Gaida et al. [10], the main challenges are a deficiency of reliable measurement technology and process monitoring in full-scale plants to provide the required data for the model parameters. Therefore, Weinrich et al. [5,6,7] developed approaches to reduce the complexity of the ADM1, for example by representing the fermentation of carbohydrates, fats, and proteins by simply three reactions. This model works in summary on a simplified stoichiometry of the model presented by Angelidaki [11,12] and uses first-order kinetics to simulate the total biogas production of the fermentation process [7]. A further example of a simplified model is the AMOCO model, described in [13], which only considered the acidogenic and methanogenic bacterial populations. Since this can only be used to model the degradation of soluble materials, a first-order hydrolysis step could be included in order for it to be applicable to the degradation of particulate materials [14].

Nevertheless, the simplified models still presuppose a number of input parameters, which primarily include the composition and degradation behavior, respectively, of the gas formation kinetics of the used substrates. Since nearly none of these parameters are available for full-scale plants, they are approximated by referring to data from experiments under laboratory conditions and digestive experiments on ruminants [7]. 

In full-scale biogas plants, the kinetics of gas formation are mainly influenced by the following parameters: the substrates used [15,16], the process temperature [17], the retention time [18], the availability of the micronutrients essential for methanogenic microorganisms [19], the ammonium nitrogen concentration in the fermentation substrate [20], and other parameters such as, for example, those described in [21]. To represent the conditions of the respective full-scale plant, all model parameters have to be carefully adjusted to the actual process state [22].

An additional approach to modeling anaerobic digestion processes is represented by data-driven models, which use, for example, artificial neural networks. An evaluation of these modeling techniques is presented in [23]. The present research investigation aims at a fundamentally different data-driven model for the simulation of biogas production in a full-scale plant, which is based on the principle of time series analysis. Using this model, correlations between time series of substrates addition and gas formation are analyzed and mathematically described. As this new model is based on time series analysis, it is self-learning and automatically adapts to the respective application, independent of the digester size and the substrates fed in.

## 2. Materials and Methods 

### 2.1. Databasis, Experimental Setup

All data used are from a full-scale biogas plant, more precisely the research biogas plant of the experimental station for agricultural science at the University of Hohenheim. The plant is operated at the location “Unterer Lindenhof” in Eningen unter Achalm, southwest Germany. The plant setup consists of two continuous stirred-tank reactors (digesters), which are covered with insulated concrete, and a secondary digester, fitted with a double membrane gas storage. Each of the three tanks has a volume of 923 m^3^. Digesters operate in the mesophilic range at 43 ± 4 °C. On average, 130 m^3^ h^−1^ biogas is produced (year 2018), with approximately 51 vol % methane (CH_4_) and 49 vol % carbon dioxide (CO_2_). The measurement of biogas production is done by a flow meter with an oscillating measuring method (hot wire sensor, company Esters Elektronik GmbH, type GD300), and is implemented separately for each digester in the biogas pipe directly at the digester outflow. The biogas is utilized via CHP-unit with an installed power of 355 kW_el_.

The supply of solid substrates is achieved via vertical mixer feeding systems, whereby both digesters have a separate feeding system. Liquid substrates are added by pumps. The quantity of substrates used is recorded by the weighing cells of the feeding system or measured via a flow meter. The hydraulic retention time (HRT) amounts to ~120 days in total. A much more detailed description of the setup of this research biogas plant can be found in Naegele and Lemmer et al. [24,25].

The data used, at an hourly resolution, the measured quantities of solid substrate (via feeding system) given in kg h^−1^ and the produced biogas quantities under standard conditions in m^3^ h^−1^. In a further investigation, the measured quantities of liquid substrates in kg h^−1^ from both digesters were taken into account. The evaluation of the model was performed using two differentiated data sets by considering both digesters as independent systems. 

All measured data are recorded and logged by the central control system of the biogas plant, and are afterwards collected, consolidated, and made available via a relational database. As a basis for the research, data from the year 2018 were used. As there were no research experiments with different feeding concepts this year, the data can be considered representative for a typical biogas plant operation.

### 2.2. Development of Process Model

All programming for visualization, modelling, and simulating was done by using the programming language R [26]. Wherever appropriate, specific packages are referenced in the following.

#### 2.2.1. Time Series Analysis

The correlations between the time series of substrate supply as the independent variable and biogas production as the dependent variable has been monitored by using the cross-correlation function. Thus, it was determined that one series related to past lags of the other series and could be used as a predictor. This analysis was done by means of the *ccf()* function of the *stats* package by P. Gilbert et al. [27]. 

#### 2.2.2. Regression Model

The basis of the developed regression model is the linear regression, expressed by the following Equation (1)
(1)$$Y_{i}=\alpha +\beta X_{i}+\varepsilon_{i}$$
whereby $Y_{i}$ is the dependent and $X_{i}$ the independent variable. The intercept is expressed as α, β is the slope of the line, and $\varepsilon_{i}$ is the error term. By including liquid manure as a second independent variable, additional $\beta_{n}$ were inserted, which are equivalently multiplied by the $X_{i}$-values. As examined by the cross-correlation function, the *X*-variables influence the *Y*-variables with time lag. Accordingly, the formula has to be adjusted with a defined number of past values of *X* as explanatory variables (2) [28].
(2)$$Y_{t}=\alpha +\beta_{0}X_{t}+\beta_{1}X_{t-1}+\ldots +\beta_{k}X_{t-k}+\;\varepsilon_{t}$$

The index *t* declares the point in time and *k* stands for the lag order. To estimate the parameters for fitting the best regression, the ordinary least square method (OLS) was used. 

For the present approach, the function *dynlm()* from the R package *dynlm* was used [29]. It employs operators to compute a lagged version of the time series in order to determine the regression model according to Equation (2).

#### 2.2.3. Simulating and Evaluation

The simulation was performed using the regression model with input data of solid substrates feeding, for which the biogas production has to be simulated, and the corresponding lagged values. 

With regard to a precise simulation of the future biogas production, the model settings had to be determined optimally. Therefore, the model was identified for (1) a suitable number of historical data (training period). Considering the model parameters with (2) a suitable number of lags, simulations can be performed over (3) a defined simulation horizon.
(1)In order to consider the most current process conditions, the training period for the model should not be too long. This ensures a simulation period with approximately the same conditions as the period of prediction of the model parameters. Therefore, the numbers of 200, 500, and 800 h were evaluated.(2)For the identification of an appropriate number of lags, it is decisive to determine in which time period the significant changes in biogas production after feeding are apparent. As described in Mauky et al. [6], feeding grass and maize silage released 62% of the total biogas production in the first 12 h after feeding. For the substrates sugar beet and crop, it is even 72%. Accordingly, the model was evaluated with a small number of 48 lags, and additionally, 72 and 96 lags.(3)A horizon of 48 h was initially defined for the simulation. In further studies, this horizon was extended. This enabled an investigation of how the simulation quality changes the further into the future the simulation is made. 


For all possible combinations of the model parameters training period, lags, and simulation horizon, the simulation model was running for every day of the year 2018 and the quality of the simulations was calculated. In total, 366 simulations were performed for each evaluation. In addition, this evaluation process was performed for the time series of digester one and digester two.

### 2.3. Quality of Simulation Model

The evaluation of simulation quality was carried out by using different accuracy parameters: the mean absolute percentage error (MAPE), the mean absolute error (MAE), and the root mean squared error (RMSE).

The *MAPE* is defined as follows (3):(3)$$MAPE=\;\frac{1}{E}\overset{E}{\underset{t=1}{\sum }}\left|\frac{z_{t}-z_{t}^{s}}{z_{t}}\right|\times 100\%$$

The *MAE* was additionally applied (4):(4)$$MAE=\;\frac{1}{E}\overset{E}{\underset{t=1}{\sum }}\left|z_{t}-z_{t}^{s}\right|$$

The third selected accuracy parameter, the *RMSE*, is defined as follows (5):(5)$$RMSE=\sqrt{\frac{1}{E}\overset{E}{\underset{t=1}{\sum }}{\left(z_{t}-z_{t}^{s}\right)}^{2}}$$

In Equations (3)–(5), *E* denotes the number of observations, $z_{t}^{s}$ is the simulated value, and $z_{t}$ corresponds to the observed value. 

For the present approach, the accuracy parameters are calculated using the R functions *mape(), mae()*, and *rmse()* from the R package *metrics* [30].

## 3. Results and Discussion

### 3.1. Results of Time Series Analysis

A first insight into solid substrate feeding and biogas production for digester one and digester two is presented in Figure 1, using data from 2018 in hourly resolution.

Digester one (Figure 1, top) was supplied with solid substrate that contained 21% solid dung, 39% maize silage, 34% grass silage, 2% whole crop silage, and 4% sugar beets. On average, 5995.34 ± 2097.37 kg of solid substrates were fed daily. Related to hourly resolution, a total of 13 feedings include a weight of more than 2000 kg h^−1^. The resulting biogas production was 70.25 ± 18.63 m^3^ h^−1^ with a total range between 2.26 m^3^ h^−1^ (except value 0) and 163.89 m^3^ h^−1^.

Digester two (Figure 1, bottom) was similarly fed with solid substrate containing 22% solid dung, 37% maize silage, 39% grass silage, and 2% whole crop silage, but no sugar beet. Here, the daily solid substrate input was 6565.23 ± 3397.46 kg d^−1^. Furthermore, the feedings show more irregularities, because altogether 28 times feeding over 2000 kg h^−1^ was registered. The mean biogas production is comparable with a value of 61.66 ± 29.71 m^3^ h^−1^. The range of biogas production is also similar, with values between 1.02 m^3^ h^−1^ (except value 0) and 160.20 m^3^ h^−1^.

The listed amounts of the various substrates show the average values for 2018. For each feeding, the composition fluctuated, depending on the availability of the substrates. For example, in regular operations, when more solid dung is available in the winter months, less maize and grass silage are used.

A significant negative correlation between solid substrate feed and biogas production was determined by cross-correlation, thus confirming the requirement to use the values of the solid substrate feed as a predictor of biogas production.

### 3.2. Regression Model and Simulating

Results of the evaluation regarding suitable settings for length of training dataset, number of lags for the model, as well as length of simulation horizon are summarized in Table 1. The evaluation was carried out over a total of 19,764 simulations by running the simulations every 24 h in 2018 and comparing them with real data.

The comparison suggests that the best results for simulating the biogas production of both digesters are achieved with a training period of 500 observations and 48 lags. These model settings indicate the best values for most accuracy parameters. For 366 simulations with a horizon of 48 h each, average MAPEs of 18.13% (digester one) and 13.87% (digester two) could be determined. Similarly, the values for RMSE and MAE are approximately 10 m^3^ h^−1^. Compared to an average biogas production of ~70 m^3^ h^−1^ (digester one) and ~60 m^3^ h^−1^ (digester two) with standard deviations of ~19 m^3^ h^−1^ and even ~30 m^3^ h^−1^, the values for RMSE and MAE are implied to be within reasonable limits. It must be mentioned that the MAE for digester two is slightly lower when using 72 lags. In general, only very small differences in the evaluation results can be recognized, which suggests that in some cases the consideration of more than 48 lags could be useful. 

Additionally, the simulation model is capable of producing simulations even over long horizons of 144 h, because the accuracy parameters show only a slight increase with a longer simulation horizon. 

As an example, the following Figure 2 shows the simulation results using a training period of 500 observations and 48 lags for digester two. 

As can be recognized in Figure 2, the simulation model is able to forecast large fluctuations in biogas production, such as those seen at the end of May 2018, with a change between 75 m^3^ h^−1^ within one day. Otherwise, as seen in November, this abrupt increase in biogas production is only reflected by the model with a slight delay. In this respect, small deviations between simulation and real values may occur. Positively noticeable is the fact that zero values of the biogas production around the month of October are represented truthfully.

In well-known models such as the ADM1, changes in biogas production are predicted by processing various parameters and process steps of digestion. However, the applied model is able to self-learn by evaluating the relationship between the time series of the substrate supply and biogas production. Due to the recurring analysis of the most recent process data, gradual shifts in feed composition are automatically considered and can be accurately represented. 

As the model always takes the individual data of a biogas plant, with the historic biogas production and feeding taken into account, the methodology is highly adaptive to a wide range of full-scale plants with their very own characteristics in terms of substrate feeding and conversion efficiency. For the presented approach, only minor or no adaptions of the methodology need to be taken into account to simulate the biogas production of other full-scale plants.

Nevertheless, further investigations and experiments regarding process failures and strong changes in feed composition would be useful to fully evaluate the extent of the reactivity of the approach. In summary, the developed model shows itself to be a very useful tool to simulate biogas production. Compared to existing simulation models, based on ADM1, only a minimum of input parameters are necessary, which can be supplied by conventional and generally implemented measurement technology. In addition to the availability of the data, it is guaranteed that data originates exclusively from the respective plant operation, and no additional assumptions based on laboratory tests are necessary. The requirements for future research projects in this area, defined by Gaida et al. [10], which include robust models in successful use in full-scale plants that are not dependent on extensive online measurement equipment and the ability to handle feeding of differentiated substrates, seems to be fulfilled with the present model.

The evaluation of the developed model also emphasized the reproducibility of the results and the adaptability to more systems by the large number of simulations (366 each), the high resolution (hourly), and the observation of two different examples of full-scale digesters (digester one and digester two). 

Going deeper into details of the model parameters allows further results of the present research investigation **.**
Figure 3 shows the boxplots of model coefficients from 366 simulations with training periods of 500 observations and 48 lags. The based time series include data from digester two.

Figure 3 illustrates the calculated coefficients as an indication of the time-dependent development of biogas production in relation to the solid substrate supply. For the purpose at hand, it could be concluded that the impact on biogas production is greatest during the first 20 h after feeding solid substrates. This emphasizes the fact that considering a larger number of lags does not improve the results of the present model. 

Further, the mapping of the median of the coefficients, visualised in Figure 3 by the red line, could indicate the course of the gas formation rate of the feeding substrates. By calculating the area under the curve, an average biogas production of 0.101 m^3^ kg^−1^ within 48 h was determined. In alignment with the substrates composition used and related average biogas yields, listed in Amon et al. [15], the calculated biogas production seems rather low. It could be assumed that one reason is the fact that only 48 h are considered and consequently less than the total gas formation potential is represented. Furthermore, it can be postulated that a residual amount of biogas is included in the model parameter intercept. Nevertheless, this evaluation could be a promising way to assess the gas formation kinetics of substrates used in full-scale plants.

A further investigation of this circumstance demonstrates the consideration of an additional regressor as a predictor on the simulation results. Besides feeding solid substrates, the dosage of liquid substrates (liquid manure) was also considered. Despite this specification of the fed substrates, the results indicated a poorer quality of the simulation. The evaluation of 366 simulations with training periods of 500 observations and 48 lags for both the solid and liquid substrates resulted in a higher MAPE of 14.92% for digester two. For comparison, the MAPE for simulations without the additional consideration of liquid manure as a second regressor is 13.87%. (cf. Table 1). Figure 4 illustrates calculated coefficients of the regressor “liquid manure” from the model.

The coefficients fluctuate around zero along all lags. In most cases, the values even fall below zero. This is due to the fact that the calculation of the coefficients depends on the intercept. The intercept is basically not interpretable, unless there is no feeding, then it can be assumed that the intercept represents the mean biogas production. Negative coefficients, as partly shown in Figure 4, indicate a higher value of the intercept, for example due to extensive feedings in the past. In addition, the supply of liquid manure in this digester proceeds at almost constant intervals and equal amounts, which could be also an explanation of the visualised coefficients. The resulting biogas production should be nearly unchanged; hence, it is not reflected in higher coefficients, but presumably included in the intercept.

The analysis of the coefficients allows for the conclusion that the feeding of liquid manure as an additional regressor has no added benefit for the simulation of biogas production in this case. It would be interesting to conduct further studies on full-scale liquid manure-based biogas plants or converting different dosages of liquid manure for the comparison of the results. 

## 4. Conclusions

The results show a fundamentally different model for the simulation of biogas production in a full-scale plant. Using time series analysis, a simulation model could be developed, which would work exclusively with historical data of biogas production and data on the amount of fed solid substrates. This offers the advantage that all necessary data can be provided via standardized measuring technology of a full-scale biogas plant. The evaluation of the model on two digesters has shown the resistance and adaptability of the model to different operating conditions. This allows for its application with nearly all full-scale plants. The quality of the simulation results achieves a mean MAPE of well below 20% in an extensive analysis of 366 simulations. The MAE and RMSE are in good range, with mean values of about 10 m^3^ h^−1^ at a mean production of about 70 and 60 m^3^ h^−1^, respectively. Remarkably, simulation quality worsened when considering liquid manure as a second regressor in the model. Once again, the simplicity and practicability of the model is underlined, as it delivers very good results with a minimum of model parameters.

In an additional investigation, it could be shown that the simulation horizon extends beyond 48 h. The simulation model is capable of producing high-quality simulations up to at least 144 h.

## Figures and Tables

**Figure 1 microorganisms-09-00324-f001:**
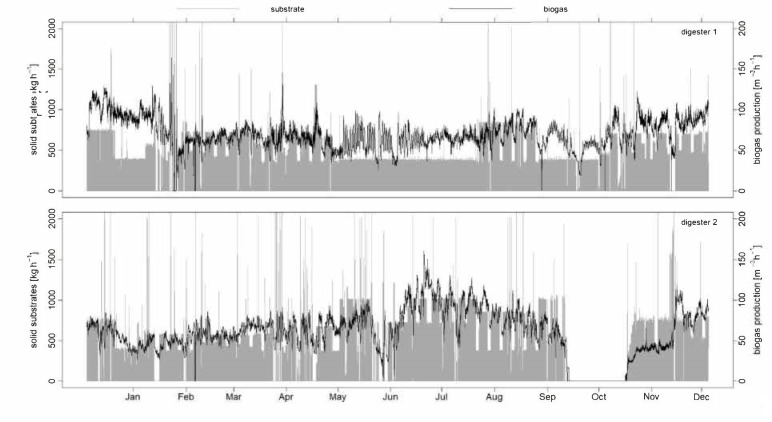
Solid substrate supply and biogas production from digester one (**top**) and digester two (**bottom**) in 2018 in hourly resolution.

**Figure 2 microorganisms-09-00324-f002:**
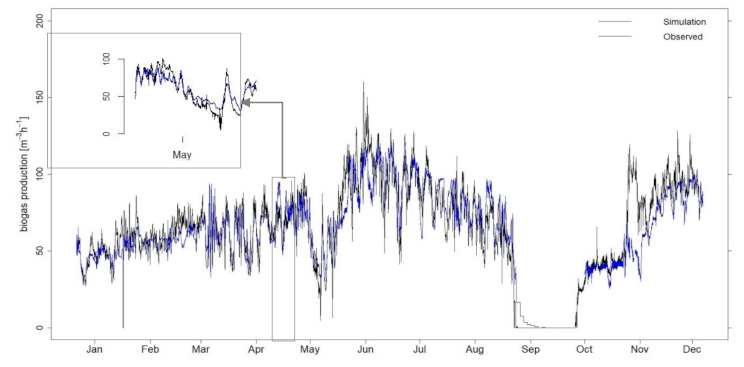
A total of 183 individual simulations, using a training period of 500 observations and 48 lags with a 48 h horizon, of biogas production for digester two, in hourly resolution, with a time window from January to December 2018. The simulated values are shown in blue, while the measured data are overlaid in black for comparison.

**Figure 3 microorganisms-09-00324-f003:**
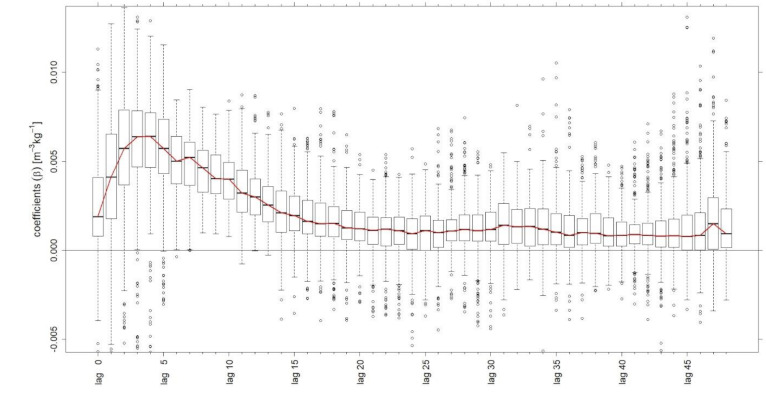
Boxplots of model coefficients β from the simulations of digester two with 500 training-observations and 48 lags.

**Figure 4 microorganisms-09-00324-f004:**
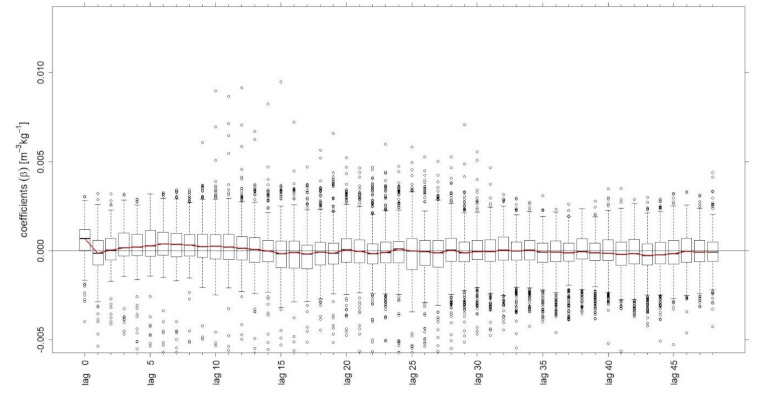
Boxplot with coefficients β of liquid substrate (liquid manure) feeding from simulation of digester two with 500 observations and 48 lags.

**Table 1 microorganisms-09-00324-t001:** Evaluation of model settings (H: horizon; T: training; L: lag) using the accuracy parameters MAPE [%], RMSE [m^3^ h^−1^], and MAE [m^3^ h^−1^] for the time series from digester one (D1) and digester two (D2). Each combination shows the mean value of 366 individual simulations.

	MAPE [%]	MAPE [%]	MAPE [%]	RMSE [m^3^ h^−1^]	RMSE [m^3^ h^−1^]	RMSE [m^3^ h^−1^]	MAE [m^3^ h^−1^]	MAE [m^3^ h^−1^]	MAE [m^3^ h^−1^]
	H: 48	H: 96	H: 144	H: 48	H: 96	H: 144	H: 48	H: 96	H: 144
T: 200	D1: 22.27	D1: 23.37	D1: 24.55	D1: 14.37	D1: 15.92	D1: 17.15	D1: 11.88	D1: 12.71	D1: 13.44
L: 48	D2: 14.05	D2: 14.78	D2: 15.23	D2: 10.50	D2: 11.36	D2: 11.87	D2: 8.98	D2: 9.53	D2: 9.85
T: 200	D1: 26.20	D1: 27.91	D1: 29.22	D1: 17.31	D1: 19.28	D1: 20.72	D1: 14.35	D1: 15.50	D1: 16.32
L: 72	D2: 15.44	D2: 16.55	D2: 16.96	D2: 11.59	D2: 12.69	D2: 13.22	D2: 9.9	D2: 10.60	D2: 10.94
T: 200	D1: 30.79	D1: 32.78	D1: 34.46	D1: 20.55	D1: 23.14	D1: 24.98	D1: 17.03	D1: 18.59	D1: 19.62
L: 96	D2: 17.09	D2: 18.43	D2: 18.68	D2: 12.79	D2: 14.02	D2: 14.49	D2: 10.80	D2: 11.59	D2: 11.91
T: 500	D1: 18.13	D1: 18.92	D1: 19.57	D1: 11.48	D1: 12.40	D1: 13.25	D1: 9.77	D1: 10.28	D1: 10.84
L: 48	D2: 13.87	D2: 14.35	D2: 14.71	D2: 10.07	D2: 10.67	D2: 11.13	D2: 8.85	D2: 9.22	D2: 9.52
T: 500	D1: 19.08	D1: 20.01	D1: 20.68	D1: 12.06	D1: 13.09	D1: 13.95	D1: 10.24	D1: 10.87	D1: 11.44
L: 72	D2: 13.88	D2: 14.37	D2: 14.64	D2: 10.09	D2: 10.71	D2: 11.13	D2: 8.81	D2: 9.19	D2: 9.46
T: 500	D1: 19.98	D1: 20.88	D1: 21.40	D1: 22.77	D1: 13.80	D1: 14.59	D1: 10.81	D1: 11.44	D1: 11.93
L: 96	D2: 14.04	D2: 14.48	D2: 14.74	D2: 10.38	D2: 11.02	D2: 11.46	D2: 9.03	D2: 9.42	D2: 9.70
T: 800	D1: 19.24	D1: 19.51	D1: 19.98	D1: 11.67	D1: 12.51	D1: 13.31	D1: 10.06	D1: 10.58	D1: 11.13
L: 48	D2: 14.77	D2: 15.05	D2: 15.26	D2: 10.77	D2: 11.46	D2: 12.02	D2: 9.64	D2: 10.06	D2: 10.46
T: 800	D1: 19.78	D1: 20.20	D1: 20.88	D1: 12.01	D1: 12.99	D1: 13.92	D1: 10.34	D1: 10.98	D1: 11.64
L: 72	D2: 14.72	D2: 15.08	D2: 15.19	D2: 10.64	D2: 11.39	D2: 11.91	D2: 9.47	D2: 9.94	D2: 10.30
T: 800	D1: 20.50	D1: 20.84	D1: 21.47	D1: 12.43	D1: 13.44	D1: 14.36	D1: 10.64	D1: 11.31	D1: 11.96
L: 96	D2: 14.77	D2: 15.12	D2: 15.26	D2: 10.77	D2: 11.55	D2: 12.09	D2: 9.57	D2: 10.06	D2: 10.44

## Data Availability

Not applicable.

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
