# Peer review of "Modeling and Simulation of Biogas Production in Full Scale with Time Series Analysis"

_microorganisms, 2021, doi:10.3390/microorganisms9020324_

Round 1

Reviewer 1 Report

This is an excellent piece of work!

The authors have provided a concise state of the art of anaerobic digestion online simulation. Their time series analysis technique is welcome.

The paper is well written up to the taste of the reviewer.

Author Response

Dear Editor and Reviewer #1, 

we are excited to be given the opportunity to receive your consistently positive feedback to our manuscript “Intelligent feeding management: Modeling and simulation of biogas production in full-scale with time series analysis”.

Thank you for your careful review of our paper. We are very pleased with your assessment that our time series analysis model advances the field of anaerobic digestion online simulation. A most welcome feedback.

Sincere regards,

Celina Dittmer

Reviewer 2 Report

The premises and motivations of the work are clearly defined as well as scientifically and technically sound, in particular the need of a model for proactive feed management.

As a further example of simplified model the AMOCO model should be also considered:

Bernard, O., Hadj-Sadok, Z., Dochain, D., Genovesi, A. and Steyer, J.-P.
Dynamical model development and parameter identification for an anaerobic wastewater treatment process, Biotechnology and Bioengineering, 2001, 75, 4, pp. 424-438.

Della Bona A., Ferretti G., Ficara E. and Malpei F.
LFT modelling and identification of anaerobic digestion, Control Engineering Practice, Volume 36, 2015, Pages 1-11.

The definition of the experimental setup is clear and complete, and the validation process is correct. The theoretical framework is also clearly explained.

The description of the identification and simulation processes should be further clarified. As far as the reviewer understand, the model is identified for A hours, considering a number of B parameters (and lags) and used to simulated the next C hours along one year, and this is replicated 366 times. An optimal choice of A, B and C is derived.

The fact that the feeding of liquid manure as an additional regressor had no added benefit is the main open question. Actually, the liquid manure gives a contribution to biogas production so neglecting it in the biogas production prediction method has no clear motivation. Some further investigation is suggested.

Author Response

Dear Editor and Reviewer #2, 

we are excited to be given the opportunity to revise our manuscript “Intelligent feeding management: Modeling and simulation of biogas production in full-scale with time series analysis”.

Thank you for your careful review of our paper and for the feedback provided. A major revision of our manuscript has been carried out to take them into account. We sincerely appreciate the suggestions you have made, and we feel that our manuscript has greatly improved as a result of your helpful comments. A list of responses to the reviewer comments is attached below.

I am looking forward to your favorable consideration

Sincere regards,

Celina Dittmer

Reviewer #2

  1. “The premises and motivations of the work are clearly defined as well as scientifically and technically sound, in particular the need of a model for proactive feed management. As a further example of simplified model the AMOCO model should be also considered:

Bernard, O., Hadj-Sadok, Z., Dochain, D., Genovesi, A. and Steyer, J.-P.

Dynamical model development and parameter identification for an anaerobic wastewater treatment process, Biotechnology and Bioengineering, 2001, 75, 4, pp. 424-438.

Della Bona A., Ferretti G., Ficara E. and Malpei F.

LFT modelling and identification of anaerobic digestion, Control Engineering Practice, Volume 36, 2015, Pages 1-11.”

Thank you for the useful comment that the AMOCO model could be mentioned in this context. I have gladly implemented this in the introduction (lines 53-58) and referred to the corresponding references.

  1. “The definition of the experimental setup is clear and complete, and the validation process is correct. The theoretical framework is also clearly explained. The description of the identification and simulation processes should be further clarified. As far as the reviewer understand, the model is identified for A hours, considering a number of B parameters (and lags) and used to simulated the next C hours along one year, and this is replicated 366 times. An optimal choice of A, B and C is derived.”

The way you understand the process is correct. I have adapted the section "Simulating and Evaluation" (lines 133-155) accordingly. In your explanations you use an understandable representation (with A, B, C), which model settings there are and how they are connected. I have followed this representation to further clarify the description.

  1. “The fact that the feeding of liquid manure as an additional regressor had no added benefit is the main open question. Actually, the liquid manure gives a contribution to biogas production so neglecting it in the biogas production prediction method has no clear motivation. Some further investigation is suggested.“

Of course, the use of liquid manure has a provable effect on biogas production. Therefore, it could be initially an unexpected result that this effect is not reflected in the calculation of the coefficients. Nevertheless, these results can be explained: In the basing digesters, the liquid manure is regularly fed in a constant amount. The biogas production attributable to the liquid manure also turns out to be constant. Therefore, no varying coefficients (corresponding to the slope) can be detected. The (constant) share of biogas production from liquid manure is reflected in the intercept of the model.

However, further studies on biogas plants that feed a changing amount of liquid manure would certainly be interesting. It can be assumed that the consideration of the additional regressor "liquid manure" is useful in such cases. In the present example, however, this could justified not be determined.

Accordingly, I have made two additions in the manuscript in lines 292-294 and 296-298.

Round 2

Reviewer 2 Report

The authors have answered to all remarks, so the paper in the opinion of the reviewer can be published.

Author Response

Dear Reviewer, 

thank you again for your careful review of our paper and the positive feedback. We are pleased that we were able to implement all your comments satisfactorily.

Sincere regards,

Celina Dittmer